# VTPerception-R1: Enhancing Multimodal Reasoning via Explicit Visual and Textual Perceptual Grounding

## Abstract

Multimodal large language models (MLLMs) often struggle to ground reasoning in perceptual evidence. We present a systematic study of perception strategies—explicit, implicit, visual, and textual—across four multimodal benchmarks and two MLLMs. Our findings show that explicit perception, especially when paired with textual cues, consistently yields the best improvements, particularly for smaller models. Based on this insight, we propose VTPerception-R1, a unified two-stage framework that decouples perception from reasoning. Stage I introduces perception-augmented fine-tuning, and Stage II applies perception-aware reinforcement learning with novel visual, textual, and consistency rewards. Experiments demonstrate that VTPerception-R1 significantly improves reasoning accuracy and robustness across diverse tasks, offering a scalable and auditable solution for perception-grounded multimodal reasoning.

## 1 Introduction

Multimodal reasoning plays a crucial role in real-world applications such as solving geometry problems and understanding complex image–text inputs. Recent developments have extended reinforcement learning with verifiable rewards (RLVR) (Dong et al., 2024) from text-only large language models (LLMs) to multimodal large language models (MLLMs). When combined with GRPO-style (Shao et al., 2024) objectives, structured format and answer rewards, and improved data or rollout strategies, these methods achieve consistent performance gains. For example, MM-Eureka (Meng et al., 2025a) and R1-VL (Zhang et al., 2025) demonstrate how RLVR can be stabilized in vision–language settings, while R1-OneVision and Vision-R1 adopt a "cold-start + RLVR" approach to bridge the modality gap and further enhance reasoning capabilities. However, these advances largely overlook a critical component: *perceptual grounding*.

Perceptual grounding, whether implicit or explicit, has been shown to influence reasoning profoundly. PAPO (Wang et al., 2025b) introduces masked-image sensitivity with regularizers to guide reasoning implicitly. In contrast, Perception-R1 (Xiao et al., 2025) employs explicit perception by leveraging annotated feedback and external LLM-based judges. Open Vision Reasoner (Wei et al., 2025) introduces a two-stage pipeline that transfers linguistic reasoning patterns to the visual domain, showing that RL can compensate for the performance degradation of cold-start visual grounding. Unfortunately, despite advances in multimodal large language models (MLLMs), existing systems remain brittle: they struggle to ground reasoning in perceptual evidence, are prone to modality bias, and lack systematic methods to unify perception and reasoning. This limits their ability to tackle tasks requiring deep understanding of complex visual-textual contexts, such as interpreting scientific figures, solving geometry problems, or answering open-ended multimodal queries. Addressing these gaps is not only scientifically compelling but also critical for building robust AI capable of reliable, explainable decision-making in real-world environments.

Other methods take a more direct route by enforcing explicit perception through model outputs. Visionary-R1 (Xia et al., 2025) imposes a "caption → reason → answer" structure, rewarding accurate captions that enable reasoning without shortcuts. Vision-SR1 (Li et al., 2025) decomposes model outputs into visual descriptions, reasoning steps, and answers, and verifies the sufficiency

of self-generated perceptions by re-prompting without images. This enables the use of self-visual rewards.

While both implicit and explicit approaches have shown benefits, most prior work lacks rigorous experimental verification to quantify the impact of visual perception—particularly in the case of explicit perception. PAPO conducts a manual audit revealing that 67% of GRPO's errors stem from perception issues, and shows that introducing an implicit perception KL reward reduces such errors by 30.5%. Vision-SR1 performs an ablation study that removes the perception self-reward, resulting in a consistent performance drop. However, neither work conducts comprehensive experiments to systematically evaluate the effectiveness of explicit visual perception on reasoning and general benchmarks. Moreover, prior research (Tourimpampa et al., 2018) highlights that both visual and textual perceptual cues jointly influence question comprehension. Yet, existing methods rarely incorporate textual perception, leading to potential over-reliance on visual inputs and increasing the risk of hallucinated reasoning.

This paper addresses these gaps through two core contributions. First, we conduct a systematic study of perception strategies for multimodal reasoning. We benchmark implicit and explicit visual and textual perception across four challenging datasets—**MMMU**, **MathVista**, **EMMA**, and a curated subset of **OlympiaBench**—using two MLLMs: **Qwen2.5-VL-32B** and **Qwen2.5-VL-7B**. We compare three grounding strategies: explicit perception notes, structured grounding via self-description, and implicit "look carefully" prompts. Our findings reveal that explicit perception consistently yields the largest performance gains, particularly for smaller models. We further show that textual perception provides meaningful improvements when capacity is limited, confirming that perceptual grounding is essential for robust multimodal reasoning and that vision and language cues jointly enhance model comprehension.

Second, we propose VTPerception-R1, a unified perception-grounded training framework that explicitly decouples seeing from reasoning. VTPerception-R1 adopts a two-stage pipeline.

**i)Stage I (Perception-augmented SFT)** trains the model to produce concise, task-relevant <description> before reasoning and answering, establishing perceptual grounding as an integral part of the reasoning process.

**ii)Stage II (Perception-aware RL)** builds upon a DAPO-style objective and introduces novel perception-specific rewards: a visual perception reward ($R_{\text{vkey}}$) to measure coverage of key visual cues, a textual perception reward ($R_{\text{tkey}}$) to measure coverage of salient textual cues, and a consistency reward ($R_{\text{cons}}$) to ensure that reasoning and answers are grounded in the perceived evidence rather than hallucinations. A perception-first weighting schedule emphasizes perceptual grounding early in training before shifting focus to correctness, producing an auditable, balanced, and end-to-end SFT→RL framework. This approach contrasts with prior methods that rely on implicit constraints or modality-specific designs, offering a principled way to jointly optimize vision and language perception for multimodal reasoning.

Through extensive experiments, we demonstrate that VTPerception-R1 significantly improves performance across diverse reasoning benchmarks, validating both the necessity of systematic perception grounding and the efficacy of our proposed framework. This work delivers the first comprehensive analysis of perception strategies for RLVR in MLLMs and provides a practical, scalable framework that bridges the gap between perception and reasoning.

This paper makes three key contributions:

*(1) Systematic evaluation of perception strategies*. We conduct the first large-scale study comparing implicit and explicit visual/textual grounding across four multimodal benchmarks—**MMMU**, **MathVista**, **EMMA**, and a curated subset of **OlympiaBench**—using **Qwen2.5-VL-32B** and **Qwen2.5-VL-7B**. Results show explicit grounding consistently delivers the largest gains, especially for smaller models, with textual perception providing complementary improvements.

*(2) VTPerception-R1*: A unified perception-grounded training framework. We propose VTPerception-R1, a two-stage framework that explicitly decouples seeing from reasoning. Stage I trains models to produce concise perceptual descriptions before reasoning, while Stage II applies perception-aware RL with novel visual, textual, and consistency rewards. A perception-first weighting schedule ensures balanced grounding and correctness.

*(3) Empirical validation and insights.* Extensive experiments demonstrate VTPerception-R1's superior reasoning performance and robustness. Our results highlight the critical role of balanced visual and textual grounding, establishing VTPerception-R1 as a principled framework for perception-grounded reasoning in MLLMs.

## 2 RELATED WORK

**Large Multimodal Reasoning Models.** Early work in multimodal reasoning combined prompt design with supervised chain-of-thought (CoT) to integrate vision and language. More recent approaches (Shao et al., 2024; Schulman et al., 2017; Liu et al., 2024) adopt reinforcement learning with verifiable rewards (RLVR), optimizing both answer correctness and structured reasoning formats under GRPO/DAPO objectives. However, directly applying text-only RLVR to vision–language tasks exposes a perception bottleneck: many reasoning failures originate in the perceptual stage, demonstrating that correctness-only objectives are insufficient.

To address this, recent works strengthen perception–reasoning coupling. PAPO (Wang et al., 2025b) uses a reverse-KL objective and entropy regularization to improve perception awareness, reducing perception-related errors. R1-style pipelines (Peng et al., 2025; Huang et al., 2025a; Tan et al., 2025; Meng et al., 2025a; Wang et al., 2025a) adapt text RLVR to MLLMs via two-stage training, reasoning control, curated datasets, and selective replay. Complementary work (Yue et al., 2025; Liu et al., 2025b;a; Yao et al., 2025; Chris et al., 2025) refines GRPO and explores rollout/data design. These approaches show that integrating perception with reasoning is more effective than optimizing correctness alone.

**Perception in Multimodal Models.** Empirical studies show correctness-driven RLVR often fails to improve perception, with perception errors dominating failure cases (Xiao et al., 2025; Lu et al., 2024; Zhang et al., 2024). Perception-targeted objectives address this: Perception-R1 (Xiao et al., 2025) rewards consistency between reasoning and visual annotations; PAPO (Wang et al., 2025b) couples perception and policy learning via reverse-KL and entropy regularization; SRPO (Wan et al., 2025) improves consolidation of visual/textual cues to strengthen reasoning. Other works treat perception as tool use (Zheng et al., 2025; Zhu et al., 2025; Su et al., 2025), enhancing external visual operations rather than intrinsic grounding. These lines show that explicit perception rewards and perception-aware objectives improve evidence coverage and reasoning quality.

## 3 SYSTEMATIC STUDY OF PERCEPTION STRATEGIES

In this section, we systematically investigate how visual and textual perception influence reasoning performance. Our experiments are designed to quantify the effectiveness of different perception strategies and their impact on downstream multimodal reasoning tasks. Settings

### 3.1 SETTINGS

To assess the influence of visual and textual perception capabilities, we conducted controlled experiments across four diverse benchmarks: **MMMU**, **MathVista**, **EMMA**, and **OlympiaBench**. These benchmarks encompass varied reasoning challenges, enabling comprehensive evaluation of perception strategies.

For visual perception, we considered three experimental conditions. In the first condition,**explicit visual perception integration**, image-based queries were enriched with structured visual perception annotations appended to the input. This allows the model to leverage explicit perceptual representations, and performance was benchmarked against these enriched inputs. The second condition, **structured visual grounding prompting**, involved engineering the system prompt to explicitly require the model to articulate its perceptual analysis of the visual information prior to producing a final reasoning output. In the third condition, **implicit visual grounding prompting**, the system prompt was minimally modified to instruct the model to "carefully observe the image" before responding, priming visual attention without requiring explicit perceptual articulation.

For textual perception, we adopted analogous conditions. In the **structured visual–textual grounding prompting** condition, the model was prompted to articulate its integrated understanding of both

| Model | Perception Setting | MathVista | MMMU | EMMA | OlympiaBench |
|---|---|---|---|---|---|
| Qwen2.5-VL-32B | Baseline | 71.11 | 60.57 | 30.85 | 16.73 |
| | Explicit visual notes | 73.33 | 62.33 | 31.97 | 18.44 |
| | Structured visual grounding | 71.39 | 61.33 | 31.71 | 16.71 |
| | Structured visual–text grounding | 71.41 | 61.28 | 31.67 | 16.76 |
| | Implicit visual grounding | 70.40 | 60.67 | 30.43 | 17.85 |
| | Implicit visual–text grounding | 70.57 | 60.42 | 30.67 | 17.88 |
| Qwen2.5-VL-7B | Baseline | 61.67 | 46.91 | 27.67 | 7.03 |
| | Explicit visual notes | 62.30 | 48.28 | 28.36 | 7.60 |
| | Structured visual grounding | 62.00 | 44.00 | 27.58 | 6.20 |
| | Structured visual–text grounding | 62.30 | 45.11 | 27.87 | 6.42 |
| | Implicit visual grounding | 61.90 | 47.67 | 27.97 | 7.20 |
| | Implicit visual–text grounding | 62.30 | 48.11 | 28.33 | 7.11 |

Table 1: Combined evaluation of Qwen2.5-VL-32B and Qwen2.5-VL-7B under visual and textual perception settings. Higher is better.

visual and textual information before producing an answer, extending structured grounding to multimodal perception. In the **implicit visual–textual grounding prompting** condition, the system prompt was lightly adjusted to instruct the model to "carefully observe the image and text", encouraging implicit multimodal attention without explicit grounding outputs.

These settings enable a rigorous, controlled comparison of perception strategies, isolating their contributions to reasoning performance across diverse tasks. For all evaluations, we use the full datasets unless otherwise specified. Specifically, for OlympiaBench, our analysis was conducted on a carefully selected subset of 600 instances, sampled randomly and proportionally across all subtasks to ensure a representative distribution of the benchmark.

### 3.2 RESULTS AND LESSONS

Table 1 shows performance differences across system prompts and explicit data integration.

**Lessons on Visual Perception**.

*(1) Qwen2.5VL-32B* Direct input augmentation with visual annotations achieves the highest overall performance, demonstrating the clear advantage of providing explicit visual understanding. Prompting the model to generate its own visual interpretation is moderately effective but generally less robust than supplying pre-processed annotations. Notably, on complex datasets such as OlympiaBench, structured prompting can be detrimental: when the model's intrinsic perceptual ability is insufficient, requiring it to articulate its perception often produces hallucinated or inaccurate observations, introducing bias and degrading reasoning performance.

*(2) Qwen2.5VL-7B* Similar to the 32B model, explicit visual annotation improves performance. However, the 7B model shows greater sensitivity to perception prompting, with structured prompts consistently reducing performance, particularly on more challenging tasks. This suggests that smaller models are more prone to self-induced perceptual errors when required to generate their own interpretations.

**Lessons on Textual Perception**.

*(1)Qwen2.5-VL-32B.* Under both explicit and implicit visual perception settings, incorporating additional textual perception yields only marginal performance gains. Moreover, the improvements are inconsistent across different benchmarks.

*(2)Qwen2.5-VL-7B.* Similar to the 32B variant, introducing textual perception on top of either explicit or implicit visual prompts results in limited performance improvement. However, the gains are notably more stable across tasks.

**Overall**. Perception prompting's impact strongly depends on model scale: larger models leverage perceptual signals more effectively, while smaller models often struggle without explicit guidance. Supplying robust perceptual information boosts performance, confirming that strong perception is

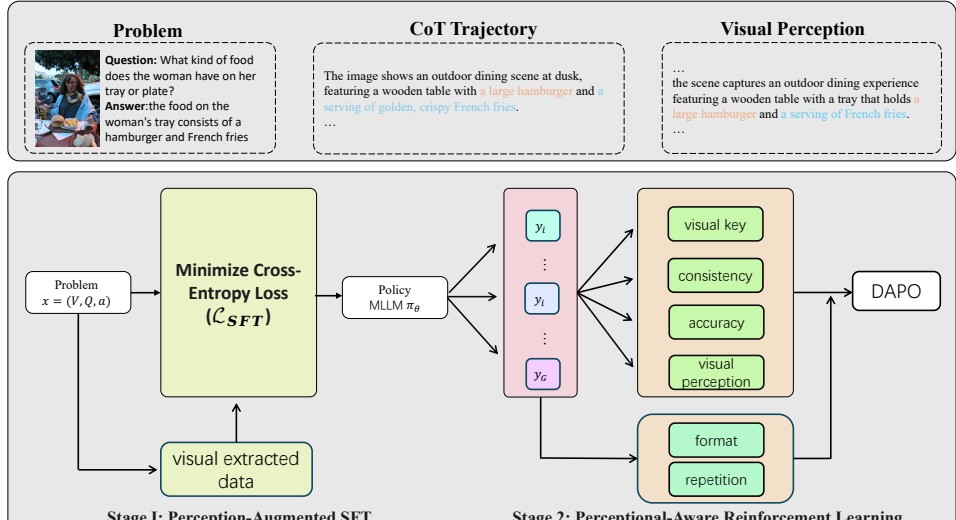

Figure 1: Overview of the proposed two-stage training pipeline VTPerception-R1. Stage 1 performs supervised fine-tuning with perception-grounded annotations, where explicit visual and textual notes are integrated into the reasoning process to strengthen multimodal perception. Stage 2 applies reinforcement learning with perception-aware rewards, further refining the Description → Think → Answer reasoning pipeline for improved consistency and interpretability.

critical for advanced reasoning. Visual perception remains a major frontier for improvement across models, whereas textual perception offers particularly high gains for smaller-scale systems.

## 4 METHOD

As discussed in Sec. 3, our analysis reveals three key observations. First, both the 32B and 7B models benefit from explicit perception prompting, with the 7B model showing greater potential for improvement. Second, the 7B model's baseline visual perception is relatively weak, requiring additional supervision to reliably ground visual information. Third, responses to textual perception prompts are unstable, indicating the need for better perception alignment.

Motivated by these insights, we propose a two-stage training pipeline, VTPerception-R1, to systematically enhance multimodal perception, particularly for the 7B variant. The overall framework is illustrated in Fig. 1.

*Stage 1*: Visual Perception–Augmented Supervised Fine-Tuning (SFT). This stage trains the model to generate focused ¡description¿ fields, improving its ability to extract and articulate visual content and grounding reasoning inputs.

*Stage 2*: Perception-Aware Reinforcement Learning (RL). Building on Stage 1, this stage refines reasoning with a composite reward that encourages (i) capturing key evidence in <description>and <think>steps, and (ii) logical consistency between evidence and answers.

This design follows two principles: (a) Make perception explicit before reasoning to improve grounding and transparency, and (b) Reward what matters for reasoning to ensure faithful, consistent outputs. Empirically, VTPerception-R1significantly boosts evidence recall, reasoning consistency, and answer quality, especially for perception-deficient smaller models.

### 4.1 STAGE I: PERCEPTION-AUGMENTED SFT

**Data and template.** Starting from LLaVA-CoT style data, we convert each instance to a structured target:

$$<description>\ldots</description><think>\ldots</think><answer>\ldots</answer>. \quad (1)$$

Here, <description>summarizes only the visual/textual evidence that is relevant to the question and useful for reasoning (not a generic caption). We retain the original chain-of-thought inside <think>and the final solution in <answer>. This explicit structuring encourages the model to first ground the input in perception, then conduct reasoning in a disentangled manner.

**Objective.** We minimize token-level cross-entropy over the full target sequence:

$$\mathcal{L}_{\text{SFT}} \;=\; -\sum_t \log \pi_\theta\big(y_t \,\big|\, x, y_{<t}\big), \tag{2}$$

which bootstraps the model to (i) attend to question-relevant regions, (ii) verbalize them succinctly in <description>, and (iii) reason over these cues. This establishes a stable perception-to-description interface that Stage II can further refine.

**Motivation.** Directly training on raw CoT-style data often entangles perception, reasoning, and answering, making it difficult for the model to learn robust visual grounding. By inserting the <description>stage, we encourage the model to first distill multimodal evidence into a compact, interpretable representation. This intermediate step reduces spurious correlations, improves generalization, and provides a controllable interface that can be explicitly inspected or modified during inference. In practice, the <description>field resembles a structured "scratchpad" for perception, while the subsequent <think>step focuses purely on abstract reasoning.

**Effect.** This stage produces a model that is able to (i) highlight salient objects and attributes, (ii) capture their spatial or semantic relations, and (iii) chain them into reasoning sequences leading to the final answer. Empirically, we find that Perception-Augmented SFT improves consistency between visual grounding and reasoning, and also simplifies downstream alignment, since the model is already trained to organize its outputs into modular components.

## 4.2 Stage II: Perception-Aware RL

**Decoupled Clip and Dynamic sAmpling Policy Optimization (DAPO).** DAPO is a reinforcement learning algorithm designed for large-scale reasoning models with long chain-of-thought outputs. Compared to earlier methods such as GRPO, it improves both stability and efficiency by introducing four key techniques: asymmetric clipping, dynamic sampling, token-level optimization, and overlong reward shaping. Formally, for each prompt $x$, $G$ responses $\{o_i\}_{i=1}^G$ are sampled, and group-relative advantages are computed to normalize rewards. The training objective is a clipped-ratio policy gradient:

$$J(\theta) \;=\; \mathbb{E}_{\{o_i\}\sim\pi_{\theta_{\text{old}}}}\left[\frac{1}{\sum_i |o_i|}\sum_{i=1}^{G}\sum_{t=1}^{|o_i|}\min\Big(r_{i,t}(\theta)\,\widehat{A}_{i,t},\; \text{clip}\big(r_{i,t}(\theta),\, 1-\varepsilon_{\text{low}},\, 1+\varepsilon_{\text{high}}\big)\widehat{A}_{i,t}\Big)\right],$$

where $r_{i,t}(\theta)$ is the token-level importance ratio and $\widehat{A}_{i,t}$ the normalized advantage. To avoid degenerate updates, DAPO excludes trivial groups where all rollouts are correct or all are wrong.

These improvements allow DAPO to provide stronger learning signals for long reasoning traces, improving convergence and reducing instability. In our work, DAPO serves as the backbone optimization method, enabling perception-aware rewards to directly shape both description quality and reasoning accuracy.

**Perception-Aware Reward.** As is described in Section A.1, we motivated the model to enhance the capability of description before think answer. We want the model to reason according to the information they have seen. What is more, the combined attention of textual information and visual information are all important. The total reward is a weighted sum:

$$R \;=\; R_{\text{acc}} \;+\; R_{\text{fmt}} \;+\; R_{\text{vkey}} \;+\; R_{\text{tkey}} \;+\; R_{\text{rep}} \;+\; R_{\text{cons}}. \tag{3}$$

We describe each component below.

**(1) Answer accuracy ($R_{\text{acc}}$).** This reward measures whether <answer> matches the ground truth, providing a direct, task-specific signal that reinforces correct reasoning and supports perception-related rewards.

**(2) Format compliance ($R_{\mathbf{fmt}}$).** This reward enforces the template <description>$\to$ <think>$\to$ <answer>, penalizing missing or repeated segments to ensure consistent, perception-grounded reasoning.

**(3) Repetition control ($R_{\mathbf{rep}}$).** This component penalizes repeated $n$-grams to discourage verbosity and promote concise, evidence-rich <description> and <think> segments.

**(4) Visual key-info ($R_{\mathbf{vkey}}$).** We enhance the data with key visual cues that are directly relevant to the task, ensuring the model captures meaningful visual information. The reward measures how well <description> covers these critical visual elements. Given a key set $K$ and the set of facts $D$ extracted from <description>, we compute the recall $\mathrm{cov} = \frac{|K \cap D|}{|K|}$ and discretize it into three levels:

$$R_{\mathrm{vkey}} = \begin{cases} 1.0, & \text{if } \mathrm{cov} \geq \tau_{\mathrm{hi}}, \\ 0.5, & \text{if } \tau_{\mathrm{lo}} \leq \mathrm{cov} < \tau_{\mathrm{hi}}, \\ 0.0, & \text{otherwise.} \end{cases} \tag{4}$$

This reward encourages the model to capture task-relevant visual evidence faithfully, thereby improving its ability to perceive and utilize visual information in downstream reasoning.

**(5) Textual key-info ($R_{\mathbf{tkey}}$).** To strengthen textual perception, we augment training data with task-specific key information—such as OCR text, numerical values, unit constraints, symbols, and relevant commonsense cues. The reward measures how well <think> incorporates these critical textual elements. Given a key set $K$ and the set of facts $D$ extracted from <think>, we compute the recall $\mathrm{cov} = \frac{|K \cap D|}{|K|}$ and discretize it into three levels:

$$R_{\mathrm{tkey}} = \begin{cases} 1.0, & \text{if } \mathrm{cov} \geq \tau_{\mathrm{hi}}, \\ 0.5, & \text{if } \tau_{\mathrm{lo}} \leq \mathrm{cov} < \tau_{\mathrm{hi}}, \\ 0.0, & \text{otherwise.} \end{cases} \tag{5}$$

As a complementary signal to $R_{\mathrm{vkey}}$, this reward ensures that reasoning does not ignore textual evidence and helps maintain a balance between visual and textual perception.

**(6) Desc–reasoning consistency ($R_{\mathbf{cons}}$).** Finally, we add a consistency reward to align perception with reasoning. It checks whether the entities, attributes, and numerical values referenced in <think> and <answer> are supported by the evidence stated in <description> and the question. Let $F_{\mathrm{ans}}$ denote the set of key items extracted from <think> + <answer>, and $E$ the items from <description> + question, defined as

$$\mathrm{cons} = \frac{|F_{\mathrm{ans}} \cap E|}{\max(1, |F_{\mathrm{ans}}|)}, \qquad R_{\mathrm{cons}} = \begin{cases} 0, & \text{if clear conflicts exist,} \\ \mathrm{cons}, & \text{otherwise.} \end{cases} \tag{6}$$

Clear conflicts (e.g., mismatched numbers or attributes) score zero. $R_{\mathrm{cons}}$ rewards alignment between reasoning and perceived evidence, reducing hallucinations and ensuring grounded answers.

# 5 EXPERIMENTS

## 5.1 SETTINGS

**Dataset.** For SFT, we sample 12K instances from LLaVA-CoT(4k) (Xu et al., 2024) and Vision-SR1(8k) (Li et al., 2025), converting each into the structured format <description>, <think>, and <answer> to train the model to verbalize task-relevant visual evidence before reasoning. For RL, we aggregate multimodal reasoning samples from MMK12(5k) (Meng et al., 2025b), LLaVA-CoT(5k), Vision-R1-rl(5k) (Huang et al., 2025b), and Mulberry(5k) (Yao et al., 2024), spanning domains such as mathematics, science, and figure comprehension. This diverse dataset supports both perception enhancement and general reasoning improvement.

**Benchmark.** We evaluate multimodal understanding and reasoning using a comprehensive suite: MMMU, MathVista, AI2D, EMMA, and Creation-MMBench.

*MMMU.* targets college-level, multi-discipline reasoning with 11.5K image–text questions across six core disciplines.

| Method | MathVista | MMMU | EMMA | AI2D | C-MMBench | C-MMBench-TO |
|---|---|---|---|---|---|---|
| Vision-SR1-7B | 71.1 | 54.9 | 28.3 | 81.0 | 39.7 | 42.4 |
| Vision-R1-7B | **71.3** | 44.9 | 27.4 | 63.2 | 52.6 | 48.5 |
| Perception-R1-7B | 67.2 | 50.9 | 27.6 | 77.4 | 41.7 | 47.9 |
| Visionary-R1 | 65.5 | 46.2 | 28.2 | 80.5 | 35.1 | 37.6 |
| MM-Eureka-Qwen-7B | 71.4 | 54.7 | 28.0 | 78.9 | 44.0 | 45.1 |
| VL-Rethinker-7B | 72.4 | **56.4** | 27.5 | 79.7 | 41.3 | 45.4 |
| Qwen2.5-VL-7B-Instruct | 66.4 | 48.4 | 28.0 | 77.2 | 43.1 | 45.0 |
| **VTPerception-R1-7B** (Before RL) | 66.4 | 50.6 | 26.6 | 80.4 | 46.7 | 47.0 |
| **VTPerception-R1-7B** | 71.0 | 52.2 | **28.8** | **82.5** | **53.2** | **50.5** |

Table 2: Performance Comparison Across Multimodal Benchmarks

*MathVista.* contains 6,141 problems drawn from 28 existing datasets plus three newly curated ones, assessing mathematical reasoning in visual contexts.

*AI2D.* evaluates diagram understanding on thousands of annotated grade-school science diagrams paired with multiple-choice questions.

*EMMA.* measures integrated cross-modal reasoning in mathematics, physics, chemistry, and coding, requiring organic image–text reasoning.

*Creation-MMBench.* specifically assesses context-aware creative capabilities of MLLMs with 765 instances over 51 fine-grained tasks and instance-specific criteria (we also report results on its text-only variant, Creation-MMBench-TO).

**Training details**. For supervised fine-tuning (SFT), we initialized from Qwen2.5-VL-7B-Instruct and tuned all parameters on a merged multimodal dataset of ∼12K samples for 3 epochs, with a learning rate of $1 \times 10^{-5}$, weight decay of 0.1, batch size 1, and gradient accumulation of 8. bf16 precision, gradient checkpointing, and DeepSpeed ZeRO-3 were enabled to support long-context multimodal inputs. For reinforcement learning (RL), we adopted a DAPO-style framework implemented in EasyR1-perc, distributed with Ray across one main node and an additional ORM node for reward computation. Tensor parallel size was set to 4. The reward function combined answer accuracy, format compliance, key visual/textual information, n-gram penalty, and consistency, with tuned weights. RL training ran on ∼22K samples for 2 epochs.

## 5.2 EVALUATION

We report performance across six representative multimodal reasoning benchmarks in Table 2. Our method (VTPerception-R1-7B), after reinforcement learning (RL), consistently outperforms its supervised fine-tuning (SFT) baseline and demonstrates competitive or superior results compared to existing strong baselines.

**Comparison with Prior Methods.** VTPerception-R1-7B achieves new state-of-the-art results on four out of six benchmarks—AI2D (82.5), Creation-MMBench (53.2), C-MMBench-TO (50.5), and EMMA (28.8). Notably, it surpasses VL-Rethinker (79.7) and MM-Eureka (78.9) on AI2D, highlighting its advantage on diagram-heavy perception tasks. On EMMA and both variants of C-MMBench, our model significantly improves over Vision-R1, Perception-R1, and MM-Eureka, demonstrating that incorporating both visual and textual perception rewards yields more reliable and grounded outputs.

**Effectiveness of Reinforcement Learning.** The RL stage contributes substantial improvements across all tasks. Compared to the SFT-only version of our model, RL brings gains of +4.6 on MathVista, +1.6 on MMMU, +2.2 on EMMA, +2.0 on AI2D, +6.5 on C-MMBench, and +3.5 on C-MMBench-TO. These gains validate not only the utility of reinforcement learning but also highlight the contribution of our method: explicitly enhancing the model's visual and textual perception capabilities. The perception-aware reward design—targeting key visual elements, textual cues, and their consistency—plays a central role, particularly in benchmarks requiring compositional reasoning and fine-grained evidence tracking. The largest improvements on C-MMBench and AI2D support our hypothesis that grounding the <description> → <think> → <answer> pipeline in concrete perceptual signals leads to more faithful, interpretable, and robust reasoning.

| Method | MathVista | MMMU | EMMA | AI2D | C-MMBench | C-MMBench-TO |
|---|---|---|---|---|---|---|
| Full model (Ours) | 65.0 | 47.9 | **26.3** | **80.8** | **44.5** | **47.9** |
| - Consistency | 64.2 | 47.1 | 26.2 | 79.6 | 41.2 | 46.2 |
| - Textual key-info | 64.3 | **49.2** | 25.6 | 80.4 | 41.9 | 44.6 |
| - Visual key-info | **66.6** | 46.7 | 26.1 | 78.8 | 43.9 | 46.5 |

Table 3: Ablation results on multiple benchmarks.

Overall, the consistent improvements across all benchmarks—particularly over perception-focused baselines like Perception-R1, Visionary-R1, and Vision-SR1—demonstrate that VTPerception-R1-7B's integration of explicit perception and structured RL rewards offers a robust and generalizable path forward for multimodal reasoning.

### 5.3 ABLATION STUDY

To demonstrate the effectiveness of our proposed method VTPerception-R1, we conducted controlled experiments under different reward configurations. For a fair comparison, all models are initialized from the same SFT checkpoint, trained with 12k samples for 3 epochs, and subsequently fine-tuned on the same RL dataset for one epoch. Performance is evaluated across multiple benchmarks, as shown in Table 3.

**Ablation Analysis.** The removal of any single reward results in a drop in overall performance, confirming the complementary roles of the three reward components. The consistency reward has the broadest impact: its absence causes the most severe declines on reasoning-intensive benchmarks (-3.26 on C-MMBench and -1.70 on C-MMBench-TO) and also reduces performance on MathVista and AI2D. This highlights the importance of enforcing a coherent $<$description$> \rightarrow <$think$> \rightarrow <$answer$>$ reasoning chain.

The text perception reward is especially critical for benchmarks like C-MMBench that rely heavily on precise textual cues, where its removal leads to drops of -2.64 and -3.31. Although a slight improvement is observed generally, likely due to reduced over-regularization, most datasets show decreased performance—indicating that weakening textual grounding tends to obscure key constraints.

The visual perception reward contributes most to diagram- or image-intensive tasks, as shown by its impact on AI2D (-2.01) and MMMU (-1.21). Interestingly, its removal slightly improves MathVista (+1.60), suggesting that certain samples may not depend strongly on fine-grained visual grounding.

In summary, the full VTPerception-R1 configuration achieves the best trade-off across all tasks, confirming the necessity of integrating all three reward signals for robust and generalizable performance.

### 6 CONCLUSION

This paper presents VTPerception-R1, a unified two-stage training framework that explicitly decouples perception from reasoning in multimodal large language models. Through a comprehensive empirical study across four challenging benchmarks, we demonstrate that explicit perception—especially when paired with textual cues—consistently improves reasoning performance, particularly for smaller models. VTPerception-R1 integrates perception-augmented supervised fine-tuning with perception-aware reinforcement learning, using novel visual, textual, and consistency rewards. Extensive experiments confirm that this perception-grounded approach not only enhances answer accuracy and robustness, but also offers a transparent and auditable path toward more reliable multimodal reasoning. Our results highlight the critical importance of grounding model reasoning in both visual and textual evidence, and pave the way for future work on integrating retrieval, tool use, and knowledge augmentation in perception-aware training pipelines.

## ETHICS STATEMENT

This work adheres to the ICLR Code of Ethics. Our study involves training and evaluating multimodal large language models (MLLMs) using publicly available datasets, such as LLaVA-CoT, MathVista, MMMU, AI2D, EMMA, and Creation-MMBench. These datasets are widely adopted in the research community and do not involve any personally identifiable information or human subject data. Our method—VTPerception-R1—aims to enhance the robustness and interpretability of multimodal reasoning by improving perceptual grounding. We believe the insights from this work are broadly beneficial and do not pose foreseeable harm or misuse. No private data, confidential material, or sensitive content was used in this study. We disclose no conflicts of interest or sponsorship bias. All authors affirm their compliance with ethical guidelines during the entire research and submission process.

## REPRODUCIBILITY STATEMENT

We have made substantial efforts to ensure the reproducibility of our results. All implementation details, including data preprocessing, model training, reward design, and evaluation protocols, are described in Sections 4and 5.1. We detail the architecture setup, hyperparameters, and training configurations for both supervised fine-tuning and reinforcement learning stages. Furthermore, we follow a deterministic, modular data cleaning pipeline for supervised data constructio in Appendix A.2, and present algorithmic pseudocode for reinforcement data distillation in Appendix A.3. All code is included as a zipped supplement in the submission. Dataset and training details are described in Section 5.1. We encourage reproducibility and further development.

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

# A APPENDIX

## A.1 MORE PRELIMINARY VALIDATION CASES AND FRAMEWORK

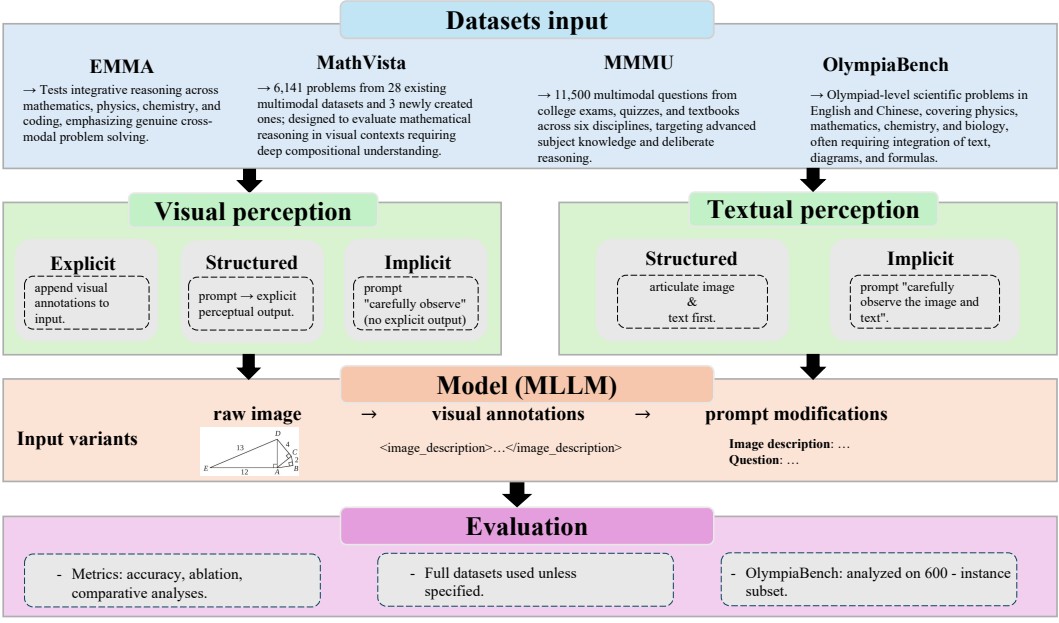

Figure 2: Verification experimental pipeline: datasets used, perception conditions (visual and textual), model input variants, and evaluation protocol.

As shown in A.1, the performance gap between models of different scales is clearly illustrated by a case study comparing Qwen2.5-VL-32B and Qwen2.5-VL-7B. When prompted for visual grounding, the 32B model showed a solid grasp of the input: it not only captured the obvious features but also inferred hidden geometric properties, correctly identifying △AOB as a right-angled triangle—an insight that guided its reasoning effectively. In contrast, the 7B model misinterpreted the diagram, hallucinating an extra property and wrongly describing △AOB as an isosceles right-angled triangle. This flawed perception disrupted its reasoning and led to an incorrect solution.

This example provides direct evidence for our central claim: the effectiveness of visual grounding prompts depends on the model's inherent perceptual ability. For a stronger model, prompting helps

**QUESTION:** In the diagram, A(0, a) lies on the y-axis above $D. If the triangles AOB and BCD have the same area, determine the value of a.

**IMAGE:**

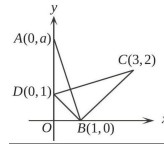

**MODEL OUTPUTS:**

The image displays a standard Cartesian coordinate system featuring two key geometric figures derived from a set of labeled points.The primary points are located on the axes: A(0, a) and D(0, 1) are on the y-axis, while B(1, 0) is on the x-axis. A fourth point, C, is located in the first quadrant at the coordinate (2, 3).These points form two distinct triangles. The first is $\triangle$ AOB, which is an isosceles right-angled triangle with its perpendicular sides lying along the x and y axes. The second figure, $\triangle$ BCD, is an obtuse triangle that connects the remaining three points. The overall composition implies a geometric relationship between the fixed area of the obtuse triangle and the variable area of the isosceles triangle.

**Qwen2.5VL-7B** ❌

The image displays two core triangles, $\triangle$ AOB and $\triangle$ BCD, on a Cartesian coordinate system.The basic geometric information shows that $\triangle$ AOB is a simple right-angled triangle whose area can be expressed directly by the variable a as $\frac{a}{2}$ However, the most critical implicit condition, revealed through coordinate analysis, is that $\triangle$ BCD is also a right-angled triangle, with its right angle at vertex B.This key hidden condition allows the area of $\triangle$ BCD to be calculated as a fixed constant, 2. Therefore, the ultimate purpose of the visual composition is to establish an algebraic equation ($\frac{a}{2}$=2) that relates a variable area to a constant area, thereby making the unknown a solvable.

**Qwen2.5VL-32B** ✅

Figure 3: Case study comparing Qwen2.5-VL-32B and Qwen2.5-VL-7B on a visual grounding task. The larger model correctly identifies △AOB as a right-angled triangle, while the smaller model mislabels it as isosceles, illustrating the role of model capacity in perception accuracy.

**QUESTION:** In the diagram, points B, P, Q, and C lie on line segment A D. The semi-circle with diameter A C has centre P and the semi-circle with diameter B D has centre Q. The two semi-circles intersect at R. If angle P R Q=40degree, determine the measure of angleA R D.

**IMAGE:**

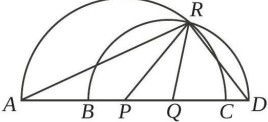

**MODEL OUTPUTS:**

The diagram is based on a straight line, AD, which contains the points A,B,P,Q,C, and D.
Above this line, there are two semi-circles.
The first semi-circle has a diameter stretching from A to C, and its center is marked as P.The second semi-circle has a diameter from B to D, with its center at Q.
The two semi-circles meet at a single point, R.
Several important triangles are formed by these points:
Triangle APR: This is an isosceles triangle because sides PA and PR are both radii of the same semi-circle. In an isosceles triangle, the angles adjacent to the base are equal, which means ∠PAR=∠APR.
Triangle DQR: This is also an isosceles triangle since QD and QR are radii. Therefore, the angles ∠QDR and ∠DQR are equal.
Triangle PRQ: This is the central triangle formed by the points P,R, and Q. The angle at vertex R, which is ∠PRQ, is given to be 40
Triangle ARD: This is the largest triangle in the figure. The angle we are interested in, ∠ARD, is the total angle at vertex R and is formed by adding up the angles ∠ARP, ∠PRQ, and ∠DRQ.

**Qwen2.5VL-7B** ❌

The diagram is constructed on a horizontal line segment, which we can label as AD. Several points, specifically A,B,P,Q,C, and D, are situated collinearly on this line.Two semi-circles are drawn on the same side of the line AD.The first, larger semi-circle is defined by the diameter AC. Its center is the point P, which is the midpoint of AC.The second semi-circle is defined by the diameter BD. Its center is the point Q, which is the midpoint of BD.These two semi-circles intersect at a point R, which is located off the line AD.
This construction forms several key triangles:
Triangle APR: This is an isosceles triangle. The sides PA and PR are both radii of the semi-circle with center P.Consequently, the base angles opposite these sides, ∠PAR and ∠ARP, are equal in measure.Triangle DQR: This is also an isosceles triangle. The sides QD and QR are both radii of the semi-circle with center Q. As a result, the base angles opposite these sides, ∠QDR and ∠DRQ, are equal in measure.Triangle PRQ: This triangle connects the two centers, P and Q, to the intersection point R. The problem states that the angle at the vertex R within this triangle, ∠PRQ, is 40 . Triangle ARD: This is the overarching triangle of interest. The angle ∠ARD at vertex R is composed of three adjacent angles: ∠ARP, ∠PRQ, and ∠DRQ. Therefore, the measure of ∠ARD is the sum of the measures of these three angles.

**Qwen2.5VL-32B** ✅

Figure 4: Caption

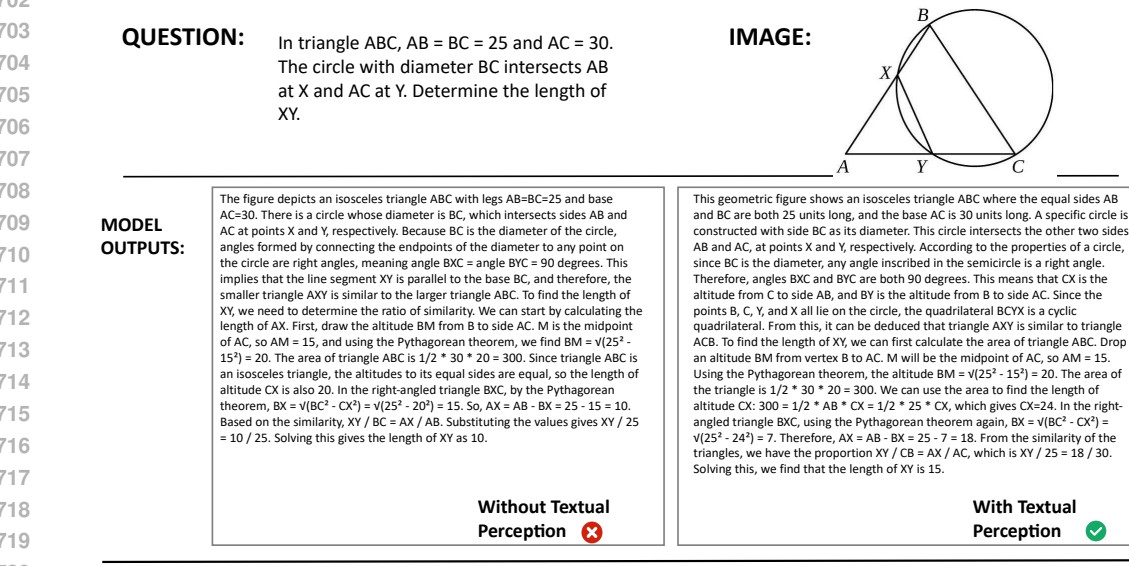

**QUESTION:** In triangle ABC, AB = BC = 25 and AC = 30. The circle with diameter BC intersects AB at X and AC at Y. Determine the length of XY.

**IMAGE:**

**MODEL OUTPUTS:**

The figure depicts an isosceles triangle ABC with legs AB=BC=25 and base AC=30. There is a circle whose diameter is BC, which intersects sides AB and AC at points X and Y, respectively. Because BC is the diameter of the circle, angles formed by connecting the endpoints of the diameter to any point on the circle are right angles, meaning angle BXC = angle BYC = 90 degrees. This implies that the line segment XY is parallel to the base BC, and therefore, the smaller triangle AXY is similar to the larger triangle ABC. To find the length of XY, we need to determine the ratio of similarity. We can start by calculating the length of AX. First, draw the altitude BM from B to side AC. M is the midpoint of AC, so AM = 15, and using the Pythagorean theorem, we find BM = √(25² - 15²) = 20. The area of triangle ABC is 1/2 * 30 * 20 = 300. Since triangle ABC is an isosceles triangle, the altitudes to its equal sides are equal, so the length of altitude CX is also 20. In the right-angled triangle BXC, by the Pythagorean theorem, BX = √(BC² - CX²) = √(25² - 20²) = 15. So, AX = AB - BX = 25 - 15 = 10. Based on the similarity, XY / BC = AX / AB. Substituting the values gives XY / 25 = 10 / 25. Solving this gives the length of XY as 10.

**Without Textual Perception** ❌

This geometric figure shows an isosceles triangle ABC where the equal sides AB and BC are both 25 units long, and the base AC is 30 units long. A specific circle is constructed with side BC as its diameter. This circle intersects the other two sides, AB and AC, at points X and Y, respectively. According to the properties of a circle, since BC is the diameter, any angle inscribed in the semicircle is a right angle. Therefore, angles BXC and BYC are both 90 degrees. This means that CX is the altitude from C to side AB, and BY is the altitude from B to side AC. Since the points B, C, Y, and X all lie on the circle, the quadrilateral BCYX is a cyclic quadrilateral. From this, it can be deduced that triangle AXY is similar to triangle ACB. To find the length of XY, we can first calculate the area of triangle ABC. Drop an altitude BM from vertex B to AC. M will be the midpoint of AC, so AM = 15. Using the Pythagorean theorem, the altitude BM = √(25² - 15²) = 20. The area of the triangle is 1/2 * 30 * 20 = 300. We can use the area to find the length of altitude CX: 300 = 1/2 * AB * CX = 1/2 * 25 * CX, which gives CX=24. In the right-angled triangle BXC, using the Pythagorean theorem again, BX = √(BC² - CX²) = √(25² - 24²) = 7. Therefore, AX = AB - BX = 25 - 7 = 18. From the similarity of the triangles, we have the proportion XY / CB = AX / AC, which is XY / 25 = 18 / 30. Solving this, we find that the length of XY is 15.

**With Textual Perception** ✅

Figure 5: Caption

unlock deeper understanding and is as effective as providing external annotations. For a weaker model, however, it can introduce errors and degrade performance.

**Verification framework (Fig. 2*case_visual1*)).** Figure 2 summarizes our verification pipeline across datasets, perception conditions, input variants, and the evaluation protocol. We benchmark two MLLMs under three visual settings—*explicit visual notes* (appending curated annotations to inputs), *structured visual grounding* (forcing a description before reasoning), and *implicit visual grounding* (light prompt to "look carefully")—and two visual–textual settings (structured vs. implicit), using full datasets except for a 600-instance, proportionally sampled subset of OlympiaBench. Accuracy is reported with ablations and comparative analyses to isolate the contribution of perception strategies. This setup operationalizes our central question: how explicit versus implicit perceptual grounding modulates downstream reasoning across tasks.

**Visual perception case (Fig. 4, *case_visual2*).** In the semicircle–intersection construction, models must decompose the figure into isosceles sub-triangles centered at the semicircle midpoints ($\triangle APR$ and $\triangle DQR$) and compose $\angle ARD$ from base angles and the given $\angle PRQ$. The larger model reliably enumerates the right substructures and completes the angle composition, whereas the smaller model tends to provide verbose but incomplete reasoning. This gap exemplifies our empirical finding: explicit visual notes help both models, but structured prompting can burden weaker perception and lead to degraded solutions—highlighting the capacity dependence of perception prompting observed in our broader study.

**Textual perception case (Fig. 5, *case_text*).** This geometry example illustrates how textual perception stabilizes reasoning. The problem asks for the length of $XY$ in an isosceles triangle $ABC$ with $AB=BC=25$ and $AC=30$, where a circle with diameter $BC$ intersects $AB$ and $AC$ at $X$ and $Y$. Without textual perception, the model introduces spurious assumptions and predicts an incorrect value; with textual perception (making the diameter–right-angle property explicit and driving similarity/power-of-point reasoning), the model recovers the correct answer $XY=15$, demonstrating that lightweight textual cues can suppress hallucinations and guide the correct derivation.

## A.2 DATASET CLEANING AND OPTIMIZATION PIPELINE FOR SFT

This appendix documents the automated cleaning and optimization pipeline used to produce a high-quality variant of the SFT dataset. The pipeline is modular and deterministic (given fixed random seeds and LLM versions), and it emphasizes (1) converting visual inputs into a single objective

representation, (2) re-deriving Chain-of-Thought (CoT) reasoning from that representation, and (3) scoring and selecting samples by multi-dimensional quality metrics.

### A.2.1 OVERVIEW

Given a raw dataset

$$\mathcal{D} = \{s_i\}_{i=1}^N, \quad s = (\text{id, image, question, original cot, original answer})$$

the pipeline transforms each sample into an enriched record:

$$\tilde{s} = (\text{id, formal\_description, cot\_thinking, final\_answer, quality\_metrics, metadata})$$

Only samples with an overall quality score above a configurable threshold $\tau$ are retained for the final cleaned dataset; an optional intelligent sampling step can further select a representative subset.

### A.2.2 STEP-BY-STEP PIPELINE

**Step 1: Image analysis and formal description generation**

The goal is a single, objective, machine-readable representation of image content (the *formal image description*).

1. VLM dense captioning: call a visual-language model (e.g., `gpt-4o` with image input or an analogous VLM) to generate a dense caption describing layout, objects, and relationships.

2. Object detection: run Grounding DINO to output object classes and bounding boxes.

3. OCR: run EasyOCR. The OCR subroutine implements error-handling and fallback strategies to avoid pipeline interruption.

4. Merging the dense caption, detection outputs and OCR text into a structured canonical text format. This description is the only image-derived input used in later CoT reconstruction to ensure objectivity.

**Step 2: Chain-of-Thought restructuring**

- Input: `question` and `formal description`.

- Prompt an LLM to produce a new reasoning trace under the explicit instruction: "Use only the provided formal image description and the question. Do not consult the original CoT or external knowledge beyond the description."

- This re-derivation corrects hallucinations in original CoTs and enforces that reasoning follows observable facts.

**Step 3: Quality assessment and data filtering**

Quality checking uses multi-dimensional metrics (1) does the formal description correctly reflect the image (consistency checks, spot-check prompts to the VLM)? (2) logical clarity and stepwise correctness of reasoning trace. (3) agreement between reasoning trace and final answer. (4) absence of unsupported claims or hallucinations. And a learned/LLM-based scorer.Return per-dimension scores in $[0, 1]$. A weighted sum can be used:

$$\text{overall\_score} = w_f s_f + w_c s_c + w_a s_a + w_m s_m,$$

with $w_f + w_c + w_a + w_m = 1$. Values of $w.$ are experiment hyperparameters (typical: $w_f = 0.30, w_c = 0.35, w_a = 0.30, w_m = 0.05$).

### A.2.3 PIPELINE PSEUDOCODE

### A.2.4 OUTPUT SCHEMA (EXAMPLE)

Each processed record is stored as JSON/JSONL with the following example structure:

---

**Algorithm 1** SFT Dataset Cleaning & Optimization Pipeline

---

**Require:** raw dataset $D$, configuration $C$
1: Initialize $\mathcal{P} \leftarrow \varnothing$ (passed set), $\mathcal{F} \leftarrow \varnothing$ (failed set)
2: **for all** sample $s$ from `SFTDataLoader`(D,C) **do**
3:     formal $\leftarrow$ ImageFormalDescriptionGenerator($s$.image, $s$.question, $C$)
4:     cot $\leftarrow$ CoTRestructurer($s$.question, formal, $C$)
5:     answer $\leftarrow$ AnswerExtractor(cot, $s$.question, $C$)
6:     metrics $\leftarrow$ QualityChecker(formal, cot, answer, $C$)
7:     **if** metrics.overall_score $\geq C$.min_score **then**
8:         add enriched record $\tilde{s}$ (including metrics) to $\mathcal{P}$
9:     **else**
10:        add $\tilde{s}$ (marked failed) to $\mathcal{F}$
11:     **end if**
12: **end for**
13: **if** C.enable_sampling **then**
14:     $\mathcal{S} \leftarrow$ DataSampler($\mathcal{P}, C$)
15:     **return** $\mathcal{S}$
16: **else**
17:     **return** $\mathcal{P}$
18: **end if**

---

```
{
  "id": "sample_0001",
  "image_path": "images/0001.jpg",
  "question": "What color is the mug on the left?",
  "formal_description": "<structured text listing objects, positions, OCR, etc.>",
  "cot_thinking": "<reconstructed chain-of-thought>",
  "final_answer": "left mug is white",
  "quality_metrics": {
    "formal_score": 0.94,
    "cot_score": 0.88,
    "answer_score": 0.97,
    "overall_score": 0.92
  },
  "passed_quality_check": true,
  "metadata": {
    "detected_objects": [
      {"class":"mug","bbox":[x1,y1,x2,y2]}
    ],
    "ocr_text": "",
    "timestamps": {...}
  }
}
```

### A.2.5 METADATA FIELDS (TABULAR SUMMARY)

**Concluding remark.** The pipeline converts noisy multimodal CoT data into a high-quality, structured dataset suitable for training and evaluation. The modular design allows swapping or upgrading detectors, OCR backends, and LLMs while preserving the single-source-of-truth principle: all reasoning must be grounded in the generated formal description.

### A.3 DATA CONSTRUCTION FOR RL

**Method Overview.** Algorithm 2 outlines our pipeline for constructing high-quality reinforcement learning (RL) data. It involves three main steps: (i) sampling diverse chain-of-thought (CoT) trajectories from multiple strong teacher models, (ii) verifying the quality of these trajectories using a budgeted evaluation process, and (iii) extracting two types of supervision signals: *visual key info* and

Table 4: Key fields stored with each processed sample.

| Field | Description |
|---|---|
| `id` | Unique sample identifier |
| `formal_description` | Canonical, objective image description (string) |
| `cot_thinking` | Reconstructed chain-of-thought (string) |
| `final_answer` | Extracted final answer (normalized string/type) |
| `quality_metrics` | Per-dimension scores and `overall_score` |
| `passed_quality_check` | Boolean pass/fail |
| `metadata.detected_objects` | Object detector output (list) |
| `metadata.ocr_text` | OCR-extracted text (string) |
| `processing_log` | Warnings, retries, and module traces |

---

**Algorithm 2** Data construction for RL: Visual & Textual Key Info

---

**Require:** Dataset $D$ of samples $(\mathbf{x}, q, a^\star)$, teacher models $\mathcal{M}$, per-teacher sample count $N$, judge budget $B$, thresholds $(\tau_{\text{acc}}, \tau_{\text{coh}}, \tau_{\text{cons}})$
**Ensure:** Distilled set $\mathcal{P}$ with CoTs and key information
1: $\mathcal{P} \leftarrow \varnothing$
2: **for all** $(\mathbf{x}, q, a^\star) \in D$ **do**
3:      Generate $N$ CoT samples per teacher $\rightarrow \mathcal{T}$
4:      Select top-$B$ by log-probability
5:      Judge each: compute correctness $s_{\text{acc}}$, coherence $s_{\text{coh}}$
6:      Keep candidates where $s_{\text{acc}} \geq \tau_{\text{acc}}$ and $s_{\text{coh}} \geq \tau_{\text{coh}}$
7:      **if** $a^\star$ is missing **then**
8:          Filter by self-consistency
9:      **else**
10:       Remove duplicates by final answer
11:      **end if**
12:      Select best trajectory $\tilde{t}$ from verified candidates
13:      Extract visual key info $\mathcal{V}$ and textual key info $\mathcal{Z}$ from $\tilde{t}$
14:      Add $(\mathbf{x}, q, \hat{a}, \tilde{t}, \{\mathcal{V}, \mathcal{Z}\})$ to $\mathcal{P}$
15: **end for**
16: **return** $\mathcal{P}$

---

*textual key info*. Each final training instance is represented as a tuple $(\mathbf{x}, q, \hat{a}, \tilde{t}, \{\mathcal{V}, \mathcal{Z}\})$, where $\hat{a}$ is a verified answer, $\tilde{t}$ is a validated trajectory, and $\mathcal{V}, \mathcal{Z}$ are the extracted key information components used in perception-aware RL.

**Teacher Ensemble and Diversity.** For a given input item $s = (\mathbf{x}, q, a^\star)$, we query several 72B-scale teacher models using stochastic decoding to generate a diverse set of reasoning paths $\mathcal{T}$. These models produce different but plausible solutions—such as varied reasoning structures, subgoal decompositions, or interpretation strategies—while remaining anchored by accurate prior knowledge.

**Budgeted Verification.** To manage annotation cost efficiently, we first rank candidate trajectories by model log-likelihood (TOPKBYLOGPROB) and select the top $B$ for further evaluation. Each selected trajectory $t$ is scored based on: (i) *Correctness* $s_{\text{acc}}$: measured by exact or normalized answer match, or by tolerance-based comparison if $a^\star$ is numeric. (ii) *Coherence* $s_{\text{coh}}$: assessed by a forward–backward consistency checker that examines step validity, use of evidence, and logical soundness. Only candidates passing both thresholds $(\tau_{\text{acc}}, \tau_{\text{coh}})$ are retained in $\mathcal{K}$. When no ground-truth answer is available, we apply a SELFCONSISTENCYFILTER to retain trajectories that reach similar conclusions. Otherwise, we de-duplicate based on final answers. We then select a single best trajectory $\tilde{t}$ using a weighted score: $w_1 s_{\text{acc}} + w_2 s_{\text{coh}}$.

**Key Info Extraction.** From the verified trajectory $\tilde{t}$, we extract two complementary types of supervision:

- **Visual key info** $\mathcal{V}$: This includes atomic, image-grounded facts such as object attributes, visual measurements, geometric constraints (e.g., parallel lines, equal lengths), and spatial relationships. These are precise and verifiable visual elements crucial for grounded reasoning.

- **Textual key info** $\mathcal{Z}$: This includes (i) a structured parse of the question's textual facts—such as entities, quantities, conditions, and units—and (ii) an *application map* that links these facts to specific reasoning steps in $\tilde{t}$, explicitly showing how each fact is used during problem solving. This provides a fine-grained, executable understanding of the question content.

**Output and Usage in RL.**   Each training item stores the verified answer $\hat{a}$, trajectory $\tilde{t}$, and extracted key information $\mathcal{V}, \mathcal{Z}$. These are used during RL to define perception-level rewards by checking how well the model's outputs align with the key information. These are then combined with answer correctness and format-based rewards. The combination of budgeted verification and self-consistency ensures high-quality data while controlling annotation cost.

**Differences from Prior Visual-Only Pipelines.**   Unlike previous pipelines that rely solely on visual supervision, our method: (i) unifies both *visual* and *textual* perception signals, (ii) introduces a budget-aware selection strategy that reduces unnecessary annotation by filtering weak candidates early, and (iii) formalizes textual key info not just as extracted facts, but as a structured fact-to-reasoning mapping, making question constraints explicit and testable.

