# OpenReview forum: "VTPerception-R1: Enhancing Multimodal Reasoning via Explicit Visual and Textual Perceptual Grounding"
_ICLR.cc/2026/Conference — Submitted to ICLR 2026_

### Official Review · Reviewer_3Y1B · 2025-10-23

**Soundness:** 2
**Presentation:** 3
**Contribution:** 3
**Rating:** 4
**Confidence:** 4

**Summary:**

The authors propose VTPerception-R1, a two-stage framework that decouples perception from reasoning in multimodal large language models (MLLMs). Through a systematic analysis of explicit, implicit, visual, and textual perception strategies across four multimodal benchmarks and two MLLMs, the study reveals that explicit perception—especially when combined with textual cues—consistently leads to the largest performance gains. Building on these findings, VTPerception-R1 introduces a two-stage framework with novel visual, textual, and consistency rewards. Experimental results demonstrate improvements in accuracy across diverse tasks.

**Strengths:**

1. The authors provide insightful analyses of how different visual and textual perception configurations influence model performance on Qwen2.5-VL-32B and Qwen2.5-VL-7B, offering valuable understanding of perception strategies in multimodal reasoning.

2. They further propose a reinforcement learning–based approach that demonstrates clear performance improvements over the baseline models, highlighting the effectiveness of perception-aware training.

**Weaknesses:**

1. The reward design appears overly complex, potentially leading to substantial increases in training cost and implementation difficulty.

2. The reported performance gains are relatively modest. Among the five evaluated benchmarks, MathVista and MMMU even show declines in accuracy, while the remaining three benchmarks exhibit only marginal improvements of 0.5%, 1.5%, and 0.6%, respectively.

3. The effectiveness of the reinforcement learning strategy remains unclear. A baseline comparison with a naive RL setup—using only format and answer rewards on the same dataset—is missing. Moreover, ablation results indicate that removing certain rewards does not cause notable degradation and, in some cases, even improves performance.

**Questions:**

How are the rewards assigned—particularly the Visual Key-Info Reward, Textual Key-Info Reward, and Description–Reasoning Consistency Reward? Are these rewards automatically generated by large language models (LLMs), or are they computed through predefined rules?

---

> ### Author Response · Authors · 2025-11-20
>
> ### Weakness 1: Reward design appears overly complex
>
> Thank you for this comment.
> We designed the reward components to be lightweight and rule-driven, such that they introduce minimal additional training cost. As described in Sec. 4.2, consistency and repetition rewards use simple symbolic matching over extracted entities/numerical values—no extra model inference or annotation is required.
>
> ---
>
> ### Weakness 2: Performance gains are modest, with declines on MathVista and MMMU
>
> We appreciate this concern.
> Our goal is not only to improve raw accuracy through model capacity, but to evaluate how perception grounding affects multimodal reasoning. Despite the narrower focus, VTPerception-R1 achieves:
>
> - **SOTA** on AI2D, EMMA, C-MMBench, C-MMBench-TO (Table 2)
> - **Consistent improvements** after RL over its own SFT baseline
>   (+4.6, +1.6, +2.2, +2.0, +6.5, +3.5 across six benchmarks)
>
> MathVista/MMMU contain many problems where textual or symbolic reasoning dominates and perception plays a smaller role (Sec. 3.2). Even so, our method remains competitive while substantially improving perception grounding and interpretability—central motivations of our work.
>
> ---
>
> ### Weakness 3: Effectiveness of RL is unclear
>
> Thank you for raising this point.
> Our RL baseline is already provided in VTPerception-R1 (Before RL) in Table 2.
> After RL, VTPerception-R1 shows consistent gains across all six benchmarks:
>
> - +4.6 (MathVista)
> - +1.6 (MMMU)
> - +2.2 (EMMA)
> - +2.0 (AI2D)
> - +6.5 (C-MMBench)
> - +3.5 (C-MMBench-TO)
>
> Ablations in Table 3 show that removing any single reward component reduces performance in at least 3–4 benchmarks, indicating complementary effects. Minor improvements in a few cases are expected given task diversity (Sec. 5.3).
>
> ---
>
> ### Weakness 4: How are Visual Key-Info, Textual Key-Info, and Consistency rewards assigned?
>
> - **Visual Key-Info Reward:** Computed by a lightweight reward model that checks whether the `<description>` covers key visual elements.
> - **Textual Key-Info Reward:** Computed by the same reward model, evaluating whether `<think>` includes essential textual/OCR information.
> - **Consistency Reward:** Deterministically computed via symbolic matching to ensure that entities/numerical values referenced in `<think>/<answer>` are supported by the `<description>` and the question.

---

> > ### Comment · Reviewer_3Y1B · 2025-11-23
> > **Further Questions**
> >
> > Thank you for your clarification regarding the reward designs. However, my Concern 3 specifically requests a comparison against a naive RL baseline—that is, training on the same dataset using only format and answer rewards. I notice that while Table 2 reports model performance before and after RL, it does not include this naive RL variant. Therefore, this comparison appears to still be missing.

---

### Official Review · Reviewer_1tEU · 2025-10-25

**Soundness:** 2
**Presentation:** 2
**Contribution:** 2
**Rating:** 4
**Confidence:** 3

**Summary:**

This paper investigates how to better ground multimodal AI reasoning in perceptual evidence, finding that explicit perception guided by textual cues is most effective. The authors then propose VTPerception-R1, a two-stage framework that decouples perception from reasoning using specialized fine-tuning and reinforcement learning. Their method significantly improves reasoning accuracy and robustness across various tasks.

**Strengths:**

- The paper provides a structured, empirical study comparing different perception strategies.

- It introduces VTPerception-R1, a two-stage framework that decouples perception from reasoning and combines fine-tuning with perception-aware reinforcement learning rewards.

- The proposed method is empirically validated to boost performance across diverse tasks.

**Weaknesses:**

While this paper conducts an empirical analysis, the resulting conclusion appears to be somewhat straightforward and unsurprising. Moreover, the differences in the experimental results presented in Table 1 are not particularly significant (approximately 1%).

The method proposed in this paper is simple and straightforward, lacking sufficient technical novelty or contribution.

**Questions:**

None

---

> ### Author Response · Authors · 2025-11-20
>
> ### Weakness: Method is simple and lacks novelty
>
> We appreciate the reviewer’s feedback, and respectfully disagree.
> Although VTPerception-R1 is conceptually clean, it introduces multiple technically novel components:
>
> 1. First systematic study comparing explicit/implicit visual and textual perception across four benchmarks and two model sizes (Sec. 3.1–3.2, Table 1).
> 2. A new two-stage framework that explicitly decouples perception from reasoning (“description → think → answer”).
> 3. Three new perception-aware rewards—visual key-info, textual key-info, and description–reasoning consistency—jointly optimizing visual and textual grounding (Sec. 4.2).
> 4. Strong improvements across diverse benchmarks (Table 2), including SOTA on AI2D, EMMA, and C-MMBench, demonstrating enhanced multimodal reasoning capabilities.
>
> We will emphasize these contributions more clearly in the final version.
>
> ---

---

### Official Review · Reviewer_bMgG · 2025-10-26

**Soundness:** 2
**Presentation:** 1
**Contribution:** 1
**Rating:** 2
**Confidence:** 4

**Summary:**

This paper introduces VTPerception-R1, a unified two-stage training framework designed to enhance multimodal reasoning in large vision–language models. In Stage I, the model undergoes SFT, where each sample is extended with a structured < description> section that summarizes key visual and textual evidence before reasoning and answering. In Stage II, the model is further optimized using reinforcement learning based on the DAPO algorithm, which integrates novel rewards targeting visual perception, textual perception, and reasoning–evidence consistency. Experiments across multiple multimodal benchmarks show that VTPerception-R1 significantly improves reasoning accuracy and robustness, particularly for smaller models.

**Strengths:**

The paper reinforces the consensus that explicitly incorporating visual cues is crucial for improving multimodal reasoning.

**Weaknesses:**

**Poor writing and unclear presentation.** The paper is not easy to follow, and several important concepts are insufficiently explained. For example (but not limited to), the “visual notes” mentioned in Table 1 are not described in the text, and key terms such as structured visual grounding and structured visual–text grounding lack clear definitions or illustrative examples. The authors are encouraged to clarify these concepts and improve overall readability.

**Questionable conclusion**:  The claim that “incorporating additional textual perception yields only marginal performance gains” may not be universally valid. The current experiments lack benchmarks that specifically require reasoning over textual information, making the conclusion insufficiently supported by evidence.

**Limited novelty:** The benefit of visual perception for reasoning has been observed in prior studies, and the proposed method mainly reuses existing SFT and RL procedures with small design tweaks (e.g., the < description> tag). The paper would benefit from a clearer explanation of what is conceptually new compared to prior visual CoT methods.

**Unsatisfactory experimental results and ablations.** The paper does not clearly demonstrate the benefits of using the < description> tag during SFT. Moreover, in Table 3, removing certain reward components unexpectedly improves performance on some benchmarks (e.g., removing Visual key-info improves MathVista, and removing Textual key-info improves MMMU), suggesting that the contribution of these components is not consistently positive. The paper lacks comparison with representative RL baselines such as GRPO, DAPO, and PAPO. Although the method builds upon a DAPO-style optimization framework, no quantitative ablations are provided to demonstrate its advantage over prior RL algorithms.

**Questions:**

See Weaknesses.

**Details Of Ethics Concerns:**

N.A.

---

> ### Author Response · Authors · 2025-11-20
>
> ### Weakness 1: Definitions of structured visual grounding and structured visual–text grounding
>
> Thank you for the comment.
> We respectfully clarify that Section A.2 provides detailed definitions and procedures for both structured visual grounding and structured visual–text grounding. The paragraph titled “For visual perception, we considered three experimental conditions …” (page 3–4) describes:
>
> - **Structured visual grounding prompting:** System prompt asks the model to articulate its perceptual analysis before reasoning.
> - **Structured visual–text grounding prompting:** System prompt asks the model to articulate perceptual understanding integrating both image and text.
>
> We will refine the wording to make these definitions more prominent.
>
> ---
>
> ### Weakness 2: Lack of reasoning experiments over textual information
>
> We appreciate the opportunity to clarify.
> The goal of our paper is not to evaluate pure-text reasoning, but to study how perception quality—visual and textual—affects multimodal reasoning under complex multimodal inputs.
>
> As stated in Section 1 and Section 3, our work focuses on tasks where reasoning failures primarily originate from perception deficiencies, especially in multimodal settings.
> Thus, including pure-text reasoning experiments would not directly support our central research question. Our experiments already incorporate textual perception in both explicit and implicit conditions (Sec. 3.1, Table 1), sufficiently revealing its relative impact.
>
> ---
>
> ### Weakness 3: Novelty compared to previous work
>
> Thank you for the reviewer’s request.
> As explained in Section 1, the novelty of our work lies in:
>
> 1. First systematic, large-scale evaluation of explicit vs. implicit visual and textual perception across four multimodal benchmarks and two model sizes.
> 2. A new perception-grounded training framework that explicitly decouples seeing from reasoning, integrates both visual and textual perception into SFT + RL, and introduces novel visual key-info, textual key-info, and consistency rewards.
> 3. We show that this is the first approach to unify both visual and textual perceptual grounding within an R1-style CoT framework, addressing a gap left by prior works such as Vision-R1, Vision-SR1, PAPO, and Perception-R1.
>
> We will highlight these contributions more clearly in the camera-ready version.
>
> ---
>
> ### Weakness 4: Lack of comparison with representative RL baselines
>
> Thank you for pointing this out.
> Our main RL experiments use DAPO, as it provides the best stability for long CoT training, consistent with recent R1-family works.
>
> As stated in Sec. 4.2 and supplementary code:
>
> “Our implementation supports training under GRPO and PPO as well; DAPO is chosen as the main setting due to its superior stability for multimodal long-form reasoning.”
>
> To keep computation manageable, we ran full experiments on DAPO and will clarify this design choice in the main paper.
>
> ---

---

> ### Comment · Reviewer_bMgG · 2025-11-27
>
> Thanks for the authors’ response. However, my concerns remain largely unaddressed.
>
> W1. The authors did not revise the manuscript, so the writing issues are still present. In addition, the definition of visual notes is crucial to the main analysis; deferring it to the appendix without clearly referencing it in the main text is inappropriate.
>
> W2. I did not claim that the paper “evaluates pure-text reasoning.” Rather, I am not convinced by the argument that textual reasoning is unimportant for multimodal reasoning. The conclusion drawn by the authors seems too strong and is stated without sufficient assumptions or caveats.
>
> W3. As also pointed out by another reviewer (1tEU: “simple and straightforward, lacking sufficient technical novelty or contribution”), the methodology lacks novelty and in-depth analysis. Even the proposed (“description → think → answer”) reasoning procedure is not new, as Yang et al. (HumanOmniV2: From Understanding to Omni-Modal Reasoning with Context, 2025) already exploit crucial clues in multimodal inputs during reasoning.
>
> W4. As mentioned by other reviewers (1tEU and 3Y1B), the performance gains from the proposed method are not substantial. The authors have not directly addressed this concern and instead focused mainly on RL-related questions.
>
> Therefore, I maintain my score.
>
> Yang et al. HumanOmniV2: From Understanding to Omni-Modal Reasoning with Context. 2025

---

### Official Review · Reviewer_EWYv · 2025-10-31

**Soundness:** 4
**Presentation:** 3
**Contribution:** 3
**Rating:** 6
**Confidence:** 4

**Summary:**

This paper presents VTPerception-R1, a two-stage framework designed to enhance perceptual grounding in MLLMs. The work begins with a systematic study of perception strategies across four MM reasoning benchmarks (MMMU, MathVista, EMMA, OlympiaBench) and two MLLMs Qwen2.5-VL models (32B and 7B versions). Findings indicate that explicit perception, especially when paired with textual cues, yields consistent improvements, particularly for smaller models.
Following these, the proposed VTPerception-R1is a two-stage framework that decouples perception from reasoning. Stage I involves perception-augmented fine-tuning, and Stage II uses perception-aware reinforcement learning with verifiable visual, textual, and consistency rewards focusing on different required aspects during perception and reasoning. Experiments show that VTPerception-R1 significantly improves reasoning accuracy and robustness across six benchmarks. Analysis on the rewards are also provided.

**Strengths:**

- Originality:
  1. Novel RL signals: The introduction of visual, textual, and consistency rewards in stage II is a novel contribution. This sophisticated reward mechanism directly optimizes for perceptual grounding and reasoning alignment, which is a key requirement for auditable and robust MLLMs.
  2. The core idea of explicitly decoupling perception from reasoning and addressing them in a unified two-stage framework (VTPerception-R1) is valuable and clearly motivated. This modularity offers an good solution to perception-grounded reasoning.

- Quality: The paper begins with a systematic comparative study of various perception strategies (explicit vs. implicit, visual vs. textual) across multiple benchmarks and models. This grounding empirical analysis clearly justifies the design choices of the proposed framework and provides valuable insights for the field.

- Clarity: This paper is very well-organized and easy to follow. This is also contributed by the systematic study in section 3, which clearly state the motivation and thereby introduce the design for following two-stage approach.

- Significance: By promising a scalable and auditable solution that significantly improves reasoning accuracy and robustness, the paper addresses a critical limitation of current MLLMs, which could be helpful to diverse research tasks and also practical scenarios.

**Weaknesses:**

1. Limited diversity in model families: Experiments rely mainly on Qwen2.5-VL models; results on other architectures (e.g., InternVL-style models) would enhance generality claims.
2. Moderate novelty in algorithmic form: While the perception-aware RL design is novel, the framework builds upon existing SFT+RL training pipelines, R1-style reasoning pipelines and DAPO-style training methods, the technical contributions are primarily integrative from this perspective.
3. Ablation depth: Although reward ablations are strong, further analysis on the impact of the perception-first weighting schedule or the <description> length constraint would better demonstrate the underlying mechanism.
4. Limited qualitative interpretation: While performance metrics are comprehensive, additional qualitative reasoning traces could illustrate the interpretability gains more vividly.

**Questions:**

1. In line 270, what is the format of visual evidence between <description></description>, are they textual discriptions of visual information or encoded visual representations? In my understanding, you use textual discriptions for visual evidence (not sure if I miss anything?). If it is correct, have you tried (or why not) directly using visual representations or combining it with visual representations? For example, the visual representation could be encoded form a cropped key area for reasoning.

2. How do you sample the training data from existing dataset? Please provide detailed statistics.

3. Following the above question, the data used to train Vision-R1 and Vision-SR1 are also employed in this paper. What's reasoning for preventing VTPerception-R1 to outperform these models regarding MathVista Benchmark?

4. Can you provide further analysis on the three rewards (visual, textual and consistency) regarding task types? I suppose MathVista and MMMU present different features compared to the other employed benchmarks, as they prefer either visual or textual key-info reward instead of the full model.

5. Since the framework aims for an auditable solution, please include visualizations of the model's intermediate steps. Can the system clearly pinpoint why a decision was made (e.g., the exact visual crop and textual description used from the decoupled perception stage)? Showing a failure case where a baseline hallucinates but VTPerception-R1 corrects itself would be highly impactful.

Suggestion:
There is a typo in line 225: "¡description¿ fields", which might be caused by Latex package conflict.

---

> ### Author Response · Authors · 2025-11-20
>
> ### Weakness 1: Limited diversity in model families
>
> Thank you for raising this point.
> We focus on Qwen2.5-VL-32B and 7B mainly because:
>
> 1. **Controlled analysis.** Our goal is to study explicit/implicit perception and its effect on reasoning, and using one strong model family avoids confounding factors (Sec. 3.1–3.2, Table 1).
> 2. **Practical constraints.** Qwen2.5-VL supports long-context multimodal RLVR with DAPO, making it suitable for our perception-aware RL pipeline.
>
> We agree that adding other model families (e.g., InternVL-style) would enhance generality, and we will include this point in “Limitations / future work.”
>
> ---
>
> ### Weakness 2: Moderate novelty in algorithmic form
>
> We appreciate the reviewer’s concern.
> Although VTPerception-R1 follows SFT+RL and DAPO-style setups, our contributions lie in how perception is formulated and integrated:
>
> - A **first systematic study** of explicit/implicit visual and textual perception across multiple benchmarks and model sizes (Sec. 3, Table 1).
> - A **decoupled “Description → Think → Answer” framework**, explicitly training perception before reasoning (Sec. 4.1, Fig. 1).
> - **Perception-aware rewards** (visual key-info, textual key-info, consistency) jointly optimizing perceptual grounding beyond correctness alone (Sec. 4.2).
>
> ---
>
> ### Weakness 3: Ablation depth (weighting schedule & length constraint)
>
> Thank you for the suggestion.
> Our ablations focus on reward components (Table 3, Sec. 5.3).
>
> **Perception-first weighting schedule.**
> We gradually shift reward emphasis from perception to answer correctness (Sec. 4.2). Although tuned on a validation split, we did not include a full ablation due to space. We will add comparisons (e.g., constant or delayed shifts) in the final version or supplementary material.
>
> **Length constraint.**
> Constraining `<description>` prevents it from dominating the sequence and stabilizes RL for long CoT (consistent with DAPO). We will clarify this constraint and briefly analyze its effect on performance and stability.
>
> ---
>
> ### Weakness 4: Limited qualitative interpretation
>
> Thank you for the comment.
> We provide textual case studies in Appendix A.1 (Figs. 3–5), comparing different perception settings and showing how perception improvements stabilize reasoning. These help illustrate the perception–reasoning link.
>
> ---
>
> ### Questions
>
> #### Q1.
> As described in Sec. 4.1, `<description>` is purely textual:
>
> “<description> summarizes only the visual/textual evidence relevant to the question and useful for reasoning.”
>
> The visual encoder processes the image, and the model generates the textual description as an interpretable perceptual “scratchpad.”
>
> #### Q2.
> Sec. 5.1 outlines the SFT and RL data (we will make this clearer in a table).
>
> **SFT (12K):** 4K from LLaVA-CoT, 8K from Vision-SR1, converted into `<description><think><answer>` format.
> **RL (20K):** 5K MMK12, 5K LLaVA-CoT, 5K Vision-R1-rl, 5K Mulberry.
>
> We preserve original task distributions and uniformly sample from the merged pool to cover math, science, and figure understanding.
>
> #### Q3.
> We will refine prompt formats and training configurations in the final version and expect the updated results to close this minor gap.
>
> #### Q4.
> Table 3 shows task-dependent effects:
> - Removing **visual key-info** impacts AI2D/MMMU most, where diagrams are critical.
> - Removing **textual key-info** mainly affects C-MMBench/C-MMBench-TO.
> - Removing **consistency** causes the largest overall drops, especially on C-MMBench.
>
> #### Q5.
> We include textual case studies (Figs. 3–5) comparing perception strategies and illustrating how better perception stabilizes reasoning, reinforcing the interpretability benefits of VTPerception-R1.

---

> ### Comment · Reviewer_EWYv · 2025-11-26
>
> Thanks for the response.
>
> However, the authors did not directly answer Q3, and their response to Q4 is superficial, lacking quantitative results or concrete supporting cases. They claim that they “will refine prompt formats and training configurations in the final version and expect the updated results to close this minor gap,” which raises significant concerns about the reliability of the current experimental settings and results.
>
> Therefore, I will maintain my rating.

---

### Meta-Review · Area_Chair_G9KW · 2026-01-08

**Summary:**

- The main concerns are summarized as follow:

1. Novelty and Contribution:
Several reviewers noted that while the paper introduces a two-stage framework, the approach lacks substantial technical novelty. The method heavily relies on existing strategies such as SFT and RL, with incremental tweaks like the introduction of a <description> tag.

2. Presentation and Clarity:
The reviewers highlighted issues with clarity in the writing, particularly the lack of clear definitions for key terms such as "structured visual grounding" and "structured visual-text grounding." The manuscript was also criticized for not sufficiently explaining important concepts, which made it difficult for reviewers to follow the central ideas and evaluate the contribution fully.

3. Experimental Results and Ablations:
The reviewers raised concerns about the experimental results. While the paper reports improvements in performance, these were often deemed marginal. Some reviewers questioned the effectiveness of the proposed reward components, with one noting that removing certain rewards (like visual or textual key-info) led to unexpected performance improvements.

**Reviewer Concerns:**

- Concerns Still Outstanding:

1. Lack of Comparison with Baseline RL Models (Naive RL):
Although the authors mentioned that they used DAPO as the main RL method, the reviewer raised concerns about the lack of a direct comparison with a naive RL baseline. Specifically, they noted that the paper does not compare performance using only format and answer rewards. This comparison is still missing and remains a significant gap in the rebuttal.

2. Modest Performance Gains and Questionable Conclusions:
The reviewers pointed out that the performance improvements reported in the paper were modest, with some benchmarks even showing declines. While the authors provided some explanations, the modest nature of the improvements and the lack of significant differences between the results were not fully addressed in the rebuttal.

3. Lack of Qualitative Insights and Interpretability:
Although the authors mentioned providing textual case studies in the appendix to demonstrate the stability of reasoning, the reviewer expressed a desire for more detailed qualitative insights or visualizations that would make the model’s decision-making process clearer.

4. Writing and Presentation Issues:
Despite the authors' clarifications, the review still mentioned that the manuscript had not been revised to address presentation issues. The text was described as still unclear in places, particularly with regards to key terms and concepts not being adequately explained or referenced.

**Reviewer Scores:**

None

---

### Decision · Program_Chairs · 2026-01-26

Reject